# Coping with chronic periprosthetic joint infection after failed revision of total knee and hip arthroplasty: a qualitative study on patient's experiences in treatment and healing

**Vincent Tilo Krenn**[1,2]\*, **Maria Sarah Bönigk**[1], **Andrej Trampuz**[1], **Martin Liebisch**[3], **Carsten Perka**[1], **Sebastian Meller**[1]

1 Center for Musculoskeletal Surgery, Charité-Universitätsmedizin Berlin, Corporate Member of Freie Universität Berlin, Humboldt-Universität zu Berlin, and Berlin Institute of Health, Berlin, Germany, 2 Empowerment Research Institute GmbH, Magdeburg, Germany, 3 Faculty of Psychology, Sigmund Freud University, Vienna, Austria

\* vincent.krenn@charite.de

## Abstract

Periprosthetic joint infections (PJI), along with the extensive medical and surgical interventions required for treatment, impose a substantial psychological burden on patients. Given the need for patients to adapt to long-term physical limitations and ongoing medical challenges, this qualitative study aims to explore the nature of psychological coping amongst patients with chronic cases of PJI. A total of 18 patients (8 men and 10 women, aged 55 to 92) who underwent a total knee or hip arthroplasty revision due to chronic PJI were recruited at a single academic institution between August 2022 and July 2023. Semi-structured interviews were conducted at two timepoints and analyzed using thematic analysis. We identified the nature of coping with PJI as patients' process-orientation towards healing. This encapsulates constant adaptation to challenges and losses in day-to-day life, managing expectations and proactively developing a sense of self-efficacy during treatment and healing. Patients expressed uncertainty and unpredictability to treatment trajectories and getting a feeling of being stuck, where health and well-being oscillates between progression and regression. These experiences contributed to unstable relationships with healthcare practitioners which were influenced by loss of trust and perceived treatment failures. A holistic view on patients, taking them seriously in their concerns and providing clear information were identified as crucial factors in shaping a positive patient-physician relationship. Acknowledging the fluctuating nature of chronic PJI treatment, physicians should adopt a process-oriented approach that promotes step-by-step healing while maintaining a positive patient-physician relationship. Recognizing the profound psychological impact, we propose the establishment of a new subdiscipline, namely 'Psycho-Endoprosthetics', to facilitate interdisciplinary collaborations for research and practice in septic surgery.

**Data availability statement:** The data from this study involve patients with a rare condition (periprosthetic joint infection). This includes potentially identifying and sensitive patient information. Due to the detailed medical histories disclosed in the study, there is an increased risk of re-identification. Anonymized data from this study will be made available to bona fide researchers after approval from the Ethics Committee of Charité. This data will be handed over to the Research Data Management of Charité to ensure persistent and long-term data storage and availability. Researchers interested in accessing the data should contact the Ethics Committee at ethikkommission@charite.de. Data-sharing requests will be assessed on a case-by-case basis following the guidelines set by the Ethics Committee to ensure participant confidentiality.

**Funding:** The author(s) received no specific funding for this work.

**Competing interests:** The authors have declared that no competing interests exist.

## Introduction

Total joint arthroplasties (TJA) of the hip and knee have markedly improved quality of life for millions by alleviating pain, restoring physical function and autonomy. In Germany, approximately 230,000 hip and 170,000 knee TJAs were performed in 2016, with an expected increase of 23% for hip and 45% for knee replacements by 2040 [1]. In the United States, projections anticipate an increase of 284% in hip and 401% in knee TJAs [2]. Despite the success of these procedures, periprosthetic joint infections (PJI) remain a rare but severe complication, often resulting in the removal of the implant, persistent pain, immobility, prolonged hospital stays, and the need for intensive antibiotic treatments with substantial side effects [3]. The incidence of PJI ranges from 0.5% to 2.6% for primary TJAs, presenting a lifelong risk of hematogenous infection at 0.07% per prosthesis per year, and is responsible for 15% to 25% of all revision surgeries [4]. While diagnostic and treatment strategies for PJI have advanced considerably [5], the post-treatment patient outcomes remain uncertain, especially due to the unclear length of recovery and the variability in regaining post-operative quality of life. Therefore, patients not only grapple with physical limitations but also endure significant psychological challenges [3,6–8] and sometimes face permanent disabilities [6,7]. Despite interdisciplinary advances in the clinical care of PJI, the psychosocial impact of PJI, particularly in chronic cases, is often underemphasized by treating physicians and remains not fully understood. Research has primarily focused on the immediate medical responses and short-term recovery processes of patients who have undergone one- or two stage revision surgery [8,9], largely overlooking the long-term psychological adjustments patients must make in order to integrate physical, mental and social consequences of chronic PJI into their lives —a highly individual process that can span over years and may be prolonged due to reinfection. Further studies on the burden of PJI have examined the implications for mental health, assessing outcome measures like depression, anxiety, and overall psychological distress [3,6]. However, there remains a significant gap in understanding the coping strategies that patients develop to manage the ongoing challenges posed by chronic PJI. This study addresses this gap by exploring the coping mechanisms that patients with chronic PJI employ to navigate their daily lives and manage the extensive repercussions of their condition. By deploying a qualitative approach, this research aims to deepen our understanding of the nuanced ways in which patients adapt to the complexities of living with a chronic PJI. Insights gleaned from this study are expected to inform better clinical practices and improve support systems, ultimately enhancing the well-being and quality of life for patients suffering from this debilitating condition. The overarching research question is stated as: What is the nature of coping with chronic PJI within the context of treatment and healing?

## Materials and methods

Following ethics board approval (EA4/040/14), we conducted a qualitative interview study including 18 patients who underwent revision surgery after the diagnosis of PJI at our specialized department. Data acquisition was performed once a week in a private treatment room at our specialized department in a single academic institution between August 2022 and October 2023. Patients were included consecutively after they provided informed consent to share their treatment and healing experiences with PJI.

Each participant was diagnosed by our infectious team, which applied the modified EBJIS criteria for diagnosing PJI [10]. PJIs were classified into high-grade and low-grade categories based on the virulence of the primary pathogen isolated. PJIs were categorized as polymicrobial if multiple pathogens were isolated from tissue cultures. Our protocol for microbiological culture involved obtaining five separate tissue samples from the deep surgical site and culturing

the supernatant after sonication of the infected implants. An additional tissue sample from the affected peri-implant site was sent for histopathological examination, which was analyzed to verify and further assess the stratification into low- and high-virulence microbial pathogens [11,12].

First-stage procedure entailed thorough debridement and implant removal. Afterwards, all patients received targeted (pathogen-specific) antibiotic therapy averaging a period of six weeks, initially intravenous in the hospital, followed by oral after discharge. The determined interval before re-implantation was based on various factors such as the physical examination, overall infectious history, condition of the wound as well as laboratory and histopathological findings. Additional debridements carried out during this period were accordingly documented. During second-stage surgery, antibiotic treatment was tailored to the identified microbial pathogens [11]. Diagnosis and management of reinfections followed the above mentioned criteria [10], and were further aligned with the Delphi consensus criteria for PJI treatment failure [12].

Inclusion criteria encompassed patients with chronic PJIs following initial total knee or hip arthroplasty that were unsuccessfully treated beforehand with either one- or two stage revision surgery. Additionally, patients who had previously undergone one- or two-stage revision surgeries for septic or aseptic causes at external facilities, were also included. Exclusion criteria were termed as: (1) acute PJIs with early postoperative onset, (2) patients who underwent part of their two-stage revision at an extern institution, (3) patients who either died or explicitly pulled out of the study, (4) patients with a critical health status that were either physically or mentally not able to participate, (5) patients under 18 years of age, (6) patients that were unable to provide informed consent.

Semi-structured interviews were conducted with a flexible approach to the interview guide (see S1 File), providing an overarching framework for predefined topics while allowing patients to address subjects personally significant to them. To do justice to the diverse treatment experiences patients with PJI had, the first interview guide focused on questions concerning how to overcome challenges, what support was needed and what expectations they had throughout the treatment journey. Since patients were at differing stages of treatment at the time point of the interview, the questions concerning challenges, support and expectations addressed their current state, past situations, and potential future trajectories. The follow-up interview centered around changes that occurred in the meantime, how they were dealt with and how patients see themselves moving onward despite the restrictions. All interviews were conducted by researcher VK either on the spot after recruitment or later at the ward, at the patients' homes or via telephone. Verbal and written informed consent were obtained in person from each patient at the outpatient clinic before the interview and refers to the conduction, audio-recording, transcription of the interview and publication of interview extracts in an anonymized manner. Each patient had the opportunity to ask research project specific questions before and after the interview. Patients were informed that they could stop the interview at any time without any repercussions or the need to provide an explanation. If patients showed signs of substantial emotional distress or seemed overwhelmed during the interview, the interviewer offered breaks or, with the patient's consent, ended the interview early with an opportunity to reschedule. After the interview, patients were asked if they would be interested in a follow-up interview six months later. Patients received a list of local support resources including counseling services and crisis hotlines and were assured that these resources were available should they need additional support after the interview. They were also encouraged to reach out to the research team with any further questions.

A total of 18 participants (see Table 1) were initially interviewed, among whom 15 were interviewed at the follow-up six months after the initial interview (3 in person and 12 via telephone). The final sample size was determined once thematic saturation was reached within analysis.

**Table 1. Demographic data and characteristics of medical history.**

| Pseud-onym | Gen-der | Age | Loca-tion | Surgical treatment | Num-ber of surgeries | Sinus tract | Pathogen | Medical history; Status at time of 1. and 2. Interview |
|---|---|---|---|---|---|---|---|---|
| Berta | F | 81 | Knee | 3x Multistage exchange; 3x DAIR[a] | 6 | no | Strepto-cocci | PJI since 2014 and multiple revisions ex domo;<br>1. Arthrodesis since last two-stage revision in domo (2022), wheelchair dependency, reimplantation unclear;<br>2. No changes in medical status |
| Chloe | F | 80 | Hip | 2x Multistage exchange; 1x DAIR | 3 | yes | polymi-crobial | PJI since 2018 and revision ex domo;<br>1. Free of infection since last one-stage revision in domo (2021);<br>2. No changes in medical status |
| Dominik | M | 70 | Hip | 3x Multistage exchange | 3 | no | E.coli[c] | PJI since 2020 and revisions ex domo;<br>1. Recovering from last one-stage exchange in domo (1-week prior interview);<br>2. undergoing rehabilitation and recovery |
| Edgar | M | 62 | Hip | 2x Multistage exchange | 2 | no | MSSE[d] | PJI with fistula since 2017 and revision ex domo;<br>1. Recovering from one-stage revision in domo (1 week ago);<br>2. Unable to reach |
| Frida | F | 80 | Knee | 5x Multistage exchange; 4x DAIR | 9 | no | MSSA | PJI since 2016 and revisions ex domo;<br>1. Planned two-stage revision (next days);<br>2. Arthrodesis, wheelchair dependency, reimplantation unclear |
| Georg | M | 55 | Knee | 3x Multistage exchange; 1x DAIR | 4 | no | MSSE | PJI with open fistula since 2016 and revisions ex domo;<br>1. Planned two-stage revision in domo (next weeks);<br>2. Girdlestone and waiting for reimplantation, wheelchair dependency |
| Irene | F | 63 | Knee | 4x Multistange exchange; 7x DAIR | 11 | yes | polymi-crobial | PJI since 2017 and revisions ex domo;<br>1. Planned two-stage revision in domo (next days);<br>2. Girdlestone and waiting for reimplantation, wheelchair dependency |
| Jan | M | 73 | Hip | 3x Multistage exchange | 3 | no | Strepto-cocci | PJI since 2019 and revisions ex domo;<br>1. Free of infection after last one-stage revision in domo (2020);<br>2. No changes in medical status |
| Karl | M | 57 | Hip | 2x Multistage exchange; 2x DAIR | 4 | no | MRSE[e] | PJI since 2021 and revisions ex domo;<br>1. Girdlestone since last two-stage revision in domo (2022), waiting for reimplantation, wheelchair dependency;<br>2. Rehabilitation after reimplantation |
| Lars | M | 81 | Hip | 3x Multistage exchange | 3 | yes | polymi-crobial | PJI since 2018 with revision ex domo;<br>1. Free of infection since last two-stage revision in domo (2020) and permanent girdlestone, wheelchair dependency;<br>2. No changes in medical status |
| Mona | F | 71 | Hip | 3x Multistage exchange; 1x DAIR | 4 | no | MSSA | PJI since 2019 with revision ex domo (2019) and one-stage revision (2021) in domo;<br>1. Pain and fluid accumulation with PJI suspicion after last revision and recommendation of revision;<br>2. Persistent pain and recommendation of revision, patient decided against revision |
| Nico | M | 61 | Hip | 2x Multistage exchange | 2 | no | Staph. lugdun-ensis | Long history of PJI after accident in young age and last revision ex domo (2019);<br>1. PJI and planned two-stage revision (next weeks);<br>2. Recovering from revision |
| Olga | F | 81 | Knee | 2x Multistage exchange; 1x DAIR | 3 | no | MSSE | PJI since 2019 with revision ex domo and in domo two stage exchange (2020);<br>1. Free of infection and permanent arthrodesis since last revision (2020);<br>2. No changes in medical status |
| Peter | M | 72 | Hip | 3x Multistage exchange | 3 | yes | E.coli | PJI since 2020 and revision ex domo and two-stage revision in domo (2022);<br>1. Girdlestone and waiting for reimplantation, wheelchair dependency;<br>2. Unable to reach |

*(Continued)*

**Table 1.** (Continued)

| Pseud- onym | Gen- der | Age | Loca- tion | Surgical treatment | Num- ber of surgeries | Sinus tract | Pathogen | Medical history; Status at time of 1. and 2. Interview |
|---|---|---|---|---|---|---|---|---|
| Simone | F | 60 | Hip | 2x Multistage exchange | 2 | no | Staph. lugdun- ensis | PJI since 2020 with revision ex domo (2020) and one-stage in domo (2021); 1. Pain and PJI suspicion since last revision; 2. Recovering from revision in domo (2022) |
| Tina | F | 82 | Hip | 4x Multistange exchange; 3x DAIR | 7 | no | Pseudo- monas aerugi- nosa | PJI since 2017 with revisions ex domo; 1. PJI with planned two stage revision in domo (next weeks); 2. Recovering from revision |
| Ute | F | 80 | Hip | 2x Multistage exchange | 2 | no | MSSE | PJI since 2020 with revision ex domo (2020) and in domo (2022); 1. Recovering from one-stage revision (2022); 2. Free of infection |

[a]DAIR = Debridement, Antibiotics and Implant Retention.

[b]MSSA = Methicillin-Sensitive Staphylococcus Aureus.

[c]E.coli = Escherichia coli.

[d]MSSE = Methicillin-Susceptible Staphylococcus Epidermidis.

[e]MRSE = Methicillin-Resistant Staphylococcus Epidermidis.

This guarantees the robustness of the findings and maximizes the data's potential for achieving the research aim [13]. Drop-outs in the follow-up were caused in one case due to the worsening of the overall status and in two other cases due to the impossibility of reaching the participants. The whole research team deemed the six-month follow-up interval as an optimal opportunity to account for potential changes in participants' circumstances that allowed to further examine development of coping strategies. By conducting two interviews, we intended to capture patients' reflections on the healing and treatment journey by providing an opportunity to refer to an oversee able period of time. Additionally, interviews at two timepoints enabled patients to clarify and expand on their earlier responses, ensuring the data accurately captured their experiences. This also gave patients an opportunity to introduce new topics or share experiences they might have missed or felt unsure about discussing during the initial interview. Through this approach, we gained a longitudinal perspective on patients' experiences that allowed us to track changes and understand evolving patterns over time. Exploring topics more deeply in this way can lead to the discovery of more detailed and complex insights since in qualitative research the experiential narrative and thus the analytical basis is created in the dialogue between interviewer and inter- viewee. After audio-recording the interviews were pseudonymized, transcribed with the software f4transkript, anonymized and imported in the software for qualitative data-management namely MAXQDA, which was used for the coding procedure. Data collection and analysis were con- ducted simultaneously and in a circulative manner.

The methodology of thematic analysis (TA) provides the opportunity for a relativistic inductively oriented analysis. Grounded in an experiential framework we focused on ampli- fying the voices of lived experiences. TA captures both semantic and latent meanings and offers a comprehensive approach that includes descriptive and interpretative accounts of the data [14,15]. A six phase TA was undertaken to explore patterns of shared meaning across data. The six phases were applied in a distinct yet recursive manner [14]. The first step of TA involved familiarization with the data by thoroughly reading and re-reading the interviews by researchers VK, ML and MB. In the second step, we performed an inductive, data-driven coding process by assigning codes to one or two-sentence segments from the transcripts, reflecting an exploratory approach. Throughout this process, inductive codes were continually

developed until all expressions related to the lived experiences of patients with PJI were identified and labeled. VK, ML, and MB independently coded 20% of the interviews. The codes were then reviewed, compared, and discussed by the team until a consensus was reached. Subsequently, VK and ML coded the remaining 80% of the interviews separately, followed by another round of review, comparison, and discussion to reach agreement. The codes were then printed and preliminarily grouped. In the third phase, all codes were categorized by VK and MB into initial themes that encapsulated the phenomena under investigation. These themes captured significant aspects of the data by portraying patterns of shared meaning embedded within the dataset [14,15]. Phase four involved reviewing and refining the themes, which were performed by VK and MB. By revisiting the themes in relation to the context of the coded sentences provided by the transcripts, an initial thematic map (see S2 Fig) was developed, visually representing the identified themes and their interrelationships. The interview transcripts and coded segments were re-examined to ensure coherence, leading to iterative adaptations of the thematic map. After capturing all phenomena associated with chronic PJI, the focus was narrowed according to the research question. Themes were reassembled and refined within the thematic map, with repeated reviews of the interview transcripts and coded segments to ensure coherence and contextual accuracy. Afterwards, final themes and subthemes were defined and named. Subthemes were identified as key concepts within their respective overarching theme. The refinement of initial themes into final themes was based on several criteria: reflection of shared patterns across the dataset, illumination of the research topic and its contextualization, non-redundancy, sufficient meaningful data to support the theme, presence of a central organizing concept, clear boundaries of the theme (for inclusion and exclusion), coherence of the theme (avoiding overly broad themes). The importance of a theme was determined not only by its prevalence but also by its salience and significance in addressing the research question. The final themes generated in this process represent the outcome of the analysis (see Fig 1). Phase five was undertaken by researchers VK and MB and focused on the refinement, renaming, and redefinition of the final themes to enhance their clarity and informative value. Lastly, phase six encompassed the writing process, which integrated the final themes into a coherent overall narrative and connected our findings to

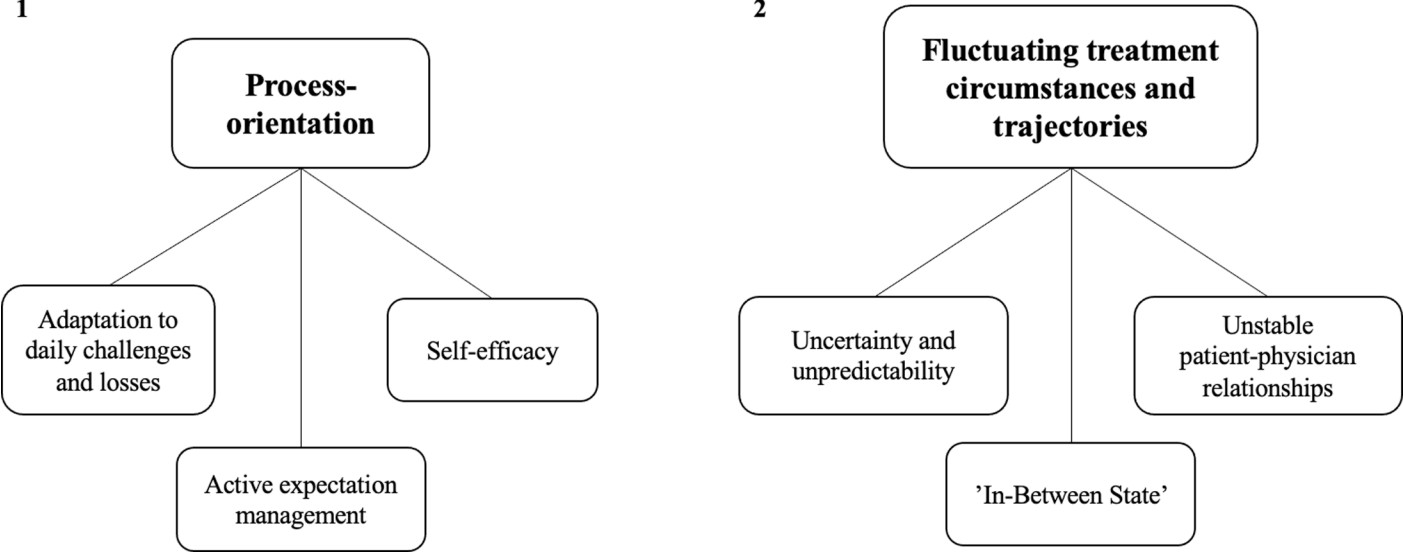

**Fig 1. Final thematic map illustrating the two main themes along with their subordinate subthemes.**

the existing literature [14]. Subsequently, lively and convincing segments from the transcripts were extracted for each theme and subtheme.

## Results

Through a detailed examination of the data, we identified key areas of analytical interest. We started by constructing a thematic map that captured all relevant aspects in the broader context of individuals' lived experiences with PJI. Focusing specifically on our research question, we analyzed how patients developed coping mechanisms to navigate challenges in day-to-day life. To contextualize these developmental processes of coping, our attention was directed towards aspects that shape the treatment experiences of PJI patients. Upon reviewing the codes, coded data segments, and the entire dataset, we shifted our attention towards the latent meanings of the expressions related to the themes. We identified a logic pattern pertaining the nature of coping mechanisms that seemed to encapsulate distinctive notions associated with managing the illness. Consequently, we formulated an analytical framework centered around two main themes: (1) process-orientation as the nature of coping with PJI, (2) the impact of fluctuating treatment circumstances and trajectories during the treatment process and respectively three subthemes within each theme (see Fig 1).

### Theme 1: Process-orientation as the nature of coping with PJI

Across our data, a process-orientated approach was crucial for patients in developing effective coping strategies to address the challenges and burdens associated with PJI. This approach reflects how the patients view both treatment and healing as an ongoing, iterative process. We identified three key elements that collectively form this process-oriented mindset: (1) Adaptation to daily challenges and losses, (2) Active expectation management, (3) Self-efficacy as a key resource for navigating treatment and healing.

**1) Adaptation to daily challenges and losses.** Patients often emphasized the profound impact of PJI on various aspects of their personal lives, leading to substantial losses. These challenges included difficulties in maintaining social connections, shaping relationships, pursuing hobbies, adhering to daily routines, finding living arrangements and maintaining mobility in public (see Table 2 for illustrative quotations). These difficulties contributed to a profound sense of disconnection from 'normal' life.

**Table 2. Disconnection from 'normal' life.**

| Dominik; 70 | So I had a structure. And this structure completely broke away, or almost completely broke away. Sport was no longer an option. I had no strength left. (1. Interview) |
|---|---|
| Edgar; 62 | Before the infection, I practiced martial arts for many years. Yes, I was in really good shape for my age. But the operation has ruined everything and I don't think I'm going to start again. (1. Interview) |
| Frida; 80 | There's always a church open on Sundays. Sometimes there are organ concerts there, but I can't always go. I have to take the bus, or someone has to take me. But that is also difficult. I've already told you. One must abstain from so many things. (2. Interview) |
| Irene; 63 | Because a lot has happened recently, so that this participation in public life is actually practically zero. (1. Interview) |
| Karl; 57 | Well, I have to admit that quite honestly and yes, it's certainly sad if you're running after your daughter now, she's seven and she says: "Daddy catch me", and I say "No, that's not. Daddy won't catch you anymore." I don't think that's positive, let's put it that way. (2. Interview) |
| Lars; 81 | That was also the reason why we had to give it (i.e., house and garden) all up and I had imagined it differently in that respect. (1. Interview) |
| Mona; 71 | And I can no longer do many of the things I used to do in my free time. For example, I used to enjoy going to the zoo or an exhibition. I can't, because after a short time the pain is so bad that I have to stop and then I just don't do it anymore. (2. Interview) |
| Simone; 60 | And then you don't have many other things to do, right? You don't get out of the house anymore. So, somewhere along the line, there's a lack of connection to normal life. (1. Interview) |

Patients across the data developed individual coping strategies to manage existential losses and adapt to changing circumstances in daily life, as presented in Table 2. Especially when confronted with the possibility of permanent disability, an active approach to restoring functionality, self-esteem and a sense of control proved crucial for initiating positive changes (see Table 3 for quotations).

**2) Active expectation management.** Patients frequently expressed the importance of a step-by-step approach to support a holistic healing process. For example, Tina (see Table 4) highlights the value of setting realistic and practical subgoals to effectively manage expectations to navigate healing and restore well-being.

The quotations in Table 4 underscore the crucial role of managing expectations in the personal healing process. Through expectation management, patients develop a sense of acceptance for the current, altered status. Tina, Lars, Mona and Frida (see Table 4) illustrate a focus on resources rather than deficits. These patients shift their focus from the overarching goal of regaining pre-infection physical functioning to preserving specific, personally significant or self-defining aspects in day-to-day life that matter most to them right now (see excerpts in Table 5).

**3) Self-efficacy as a key resource for navigating treatment and healing.** Across the data, patients develop a sense of control during their treatment and healing process, which is

**Table 3. Small changes with big impact.**

| | |
|---|---|
| Irene; 63 | But the fact that I have this very strong ambition to change that (lack of participation in public life) and therefore I can do something. I have a very large circle of friends who help me and I can draw on this. So, I can also do activities with them, mainly at home of course. (2. Interview) |
| Karl; 57 | When I was at home alone with the little one (daughter), then ((laughs)) we cooked at the kitchen table and then I pushed the pot with the crutch along the floor to the dining table. That worked wonderfully. (2. Interview) |
| Mona; 71 | I'm trying more and more to focus on positive things. Now when I watch TV, crime thrillers and these chaotic films for example, I prefer to switch and watch a nature film instead. It's interesting to see how people live in other regions. Then I am distracted. (2. Interview) |
| Nico; 61 | So, during the day, yes, small tripping hazards, I have to avoid them, I've already memorized them. So uneven floors are the worst thing there is for me. (1. Interview) |
| Tina; 82 | So a few little things, you know. So I think it's more that I've adjusted to this suffering, you know. You kind of develop a technique to deal with it. (2. Interview) |

**Table 4. Acceptance of restrictions make space for new adjustments.**

| | |
|---|---|
| Dominik; 70 | I have hope, as I said, of a healing process where I want to give myself time. At least six months to a year, I think, to get back to a certain reasonable level. So maybe I'll be able to walk five kilometers without being in pain. I don't want to go mountain climbing and bungee jumping or anything else. But a bit of a normal life. Maybe to be able to do a bit of sport again without my body immediately saying it's too much for me. (1. Interview) |
| Frida; 80 | I have a lovely balcony and I have the sun there and I treat myself to it, I always have it planted. And that's what I've done now. And then I'm happy about my flowers. Yes, you know what? You become modest. (2. Interview) |
| Karl; 57 | Well, with my first operation it went much faster. Maybe because it was now opened up three times in a row, I don't know, but it's actually slower. So, I just have to accept that, you really have to accept that it takes longer. (2. Interview) |
| Lars; 81 | And you shouldn't mourn how it used to be before. You should also forget how it used to be, not mourn it. And you should always try to keep an open mind. (2. Interview) |
| Mona; 71 | My expectations used to be much higher, and they were always fulfilled. But that's no longer the case. That's why I lower my expectations a little. If it turns out better, so much the better. (2. Interview) |
| Olga; 81 | I just expect to be able to walk reasonably well. That I will be able to go to a festivity, which I haven't had for ages. Everything was denied to me because I was only hanging around the hospital, yes. (2. Interview) |
| Tina; 82 | But it's often the case that things don't go according to plan. And then I think it's really important to say okay, it's not about being able to climb the mountain the next day, but really stage one is being able to get around on my own in a good way. (1. Interview) |

closely linked to perceptions of self-efficacy —the belief in their ability to successfully execute actions necessary to overcome challenges. Patients frequently described scenarios in which they assertively addressed challenges by actively taking charge of their situation. These actions often directly influenced both the treatment process and its outcomes. In doing so, these individuals proactively took steps towards maintaining and promoting their self-efficacy. Conversely, self-efficacy can empower patients to engage in active behaviors they believe will positively affect their situation. Therefore, we view self-efficacy as a foundational element for proactive patient participation in treatment. This is illustrated by individual quotations in Table 6, which highlight this process-oriented perspective and active role in their personal treatment and healing processes.

**Table 5. Redirecting focus from medical to personal goals.**

| | |
|---|---|
| Frida; 80 | It (surviving a septic shock caused by PJI) made me think that life still has a meaning after all. Going for a walk or with a friend, that gives you something. It's a good thing I'm not so meek and give up everything. Not that. But I have found meaning for me and I live by it without restrictions. (1. Interview) |
| Irene; 63 | I (director of a church choir) conducted from a wheelchair, but it still worked. Because everyone thinks along and everyone helps. Yes, I was able to do that too. (1. Interview) |
| Karl; 57 | Yes, I think it's very important to get these parts of your life back, perhaps to use the chance, that you can come back in a self-determined way if you are thrown backwards, so to speak. (2. Interview) |
| Lars; 81 | I can move relatively well in a wheelchair and I can basically do almost everything myself again, so I'm happy with that. (2. Interview) |
| Nico; 61 | Working in the garden is almost impossible. We have a small plot of land and it needs looking after. I always try to involve myself at least a little bit. (1. Interview) |
| Tina; 82 | But let's say, in general, what is most important is still working. That's actually what reassures me. What helps me at this moment to deal with it. (2. Interview) |
| Ute; 80 | Well, I want to carry on living the way I do now! Well, I want to have fewer operations in the future, I hope none. (2. Interview) |

**Table 6. Proactively restoring control and regaining self-efficacy.**

| | |
|---|---|
| Chloe; 80 | I simply have the opinion that you shouldn't hysterically google somewhere or do something else or get information somewhere else, but approach it calmly and objectively and see what I can take from it that could help me. But I think I've done the right thing. (1. Interview) |
| Dominik; 70 | If you want it, you have to get involved yourself and demand it. And you also get told a lot and explained a lot and then I think it's good again. But relatively little comes without demanding it. (2. Interview) |
| Edgar; 62 | And as the saying goes, hope dies last. Right? But I've now taken a proactive approach and that's a good thing! Yes, something has to happen now, because I don't want to die like this. Yes, I don't want to die like this […] so I take what the doctors do here and then I have to deliver. (1. Interview) |
| Frida; 80 | I say to myself, a doctor has also sworn an oath. And you become a doctor because you want to help people. And I don't see it that way in this case! So you have to have the courage to change doctors if it goes against the grain, right? (1. Interview) |
| Georg; 55 | Well, I think it's also important to simply determine what helps and what doesn't do you any good in situations like this. That's at least just as important. (1. Interview) |
| Irene; 63 | So what I'm actually trying to say is that I'm not trying to bury myself in my condition, but that I'm trying to be a part of it despite everything and fill my life with activities. (2. Interview) |
| Mona; 71 | Well, I don't get so upset about it all anymore. I just try to cope with the situation. And if something else (revision) is done now, then I'll take it as it comes. Ultimately, it's my decision whether or not I have the surgery. (1. Interview) |
| Simone; 60 | Well, the hardship drove me, because I was sitting at home and didn't know, so I just googled it. And infection and hip, I automatically ended up with you. So then I thought, well, at least I'll get a second opinion and then we'll have to see. Because the case has to be solved. It's not going to get any better if I put it off. (1. Interview) |
| Tina; 82 | But just imagine if I had listened to the doctors back then. Because I had reached a point where I couldn't get rid of the bacteria. The doctors said at the time: "I wouldn't have another operation if I were you. It can always go wrong. And then everything could be gone." I mean, I really struggled with myself at first. Well, I think I did the right thing. (1. Interview) |
| Ute; 80 | So at first I was totally depressed (after being advised to have another revision) and then I said "So now I'm fighting!". (1. Interview) |

The expressions of Georg, Irene, and Ute demonstrate a strong sense of self-awareness, emphasizing the importance of engaging in self-reflection during treatment. This process is crucial for recognizing personal strengths, resources, and motivators, which in turn actively enhance well-being. They engaged in anticipating individual stressors and finding ways to manage them effectively. Additionally, another source of self-efficacy and even optimism in challenging situations can be traced to the patient's past. Since many patients frequently derive a sense of self-efficacy from previous adverse life events or challenges they successfully navigated before the onset of PJI. These past experiences serve as the basis for resiliency when faced with challenges accompanied by PJI. Consider the following extracts in Table 7:

## Theme 2: Fluctuating treatment circumstances and trajectories.

Across our data, we identified a shared pattern related to the fluctuating treatment circumstances and trajectories associated with PJI. These fluctuations highlight the complex and often unpredictable nature of the disease, which significantly impacts overall disease management. To better contextualize the developmental process of coping with PJI challenges, we focused on three facets of the treatment experience that are particularly relevant for PJI patients: (1) uncertainty and unpredictability, (2) the 'In-Between State', and (3) an unstable patient-physician relationship during treatment.

**1) Uncertainty and unpredictability.** Patients often highlighted the notion of unpredictability during the treatment process, laced with uncertainty and frequent changes in recommendations for medical interventions and professional opinions. These aspects of PJI treatment exemplify how coping skills are actively developed in response, and as a protection against, the often occurring fluctuations throughout treatment and healing. Consider the following excerpts in Table 8:

Mona characterized her treatment as a trial-and-error process and expressed frustration with the fundamental lack of clear information. After receiving inconsistent recommendations from various physicians, she ultimately chose to reject further surgeries and take charge of her own treatment decisions. Similarly, Edgar described the decision-making aspect within treatment as a highly individual process influenced by a variety of factors, emphasizing the inner struggle patients may experience when deciding whether to undergo surgery. Patients frequently encounter these bifurcations along their treatment journey, necessitating decision-making that surpasses their own experiences, skills, and knowledge as laypersons. This often reinforces their sense of dependency on medical professionals in a field where treatment options and decisions can be multidirectional and unclear.

**2) 'In-Between State'.** Patients frequently describe feeling stagnation or being stuck in their treatment journey, experiencing what we characterize as an 'In-Between State', which we describe as a continuous back-and-forth movement of their perceived health status. This state

**Table 7. Mobilizing strength from critical past experiences.**

| | |
|---|---|
| Georg; 55 | I tell myself, you've done so much, achieved so much in your life. You've lost weight, the military service, your stomach operation, knee operation, you've put it all away. We're going to get that back on track too. We'll manage that, won't we? (1. Interview) |
| Mona; 71 | I always used to think to myself, when a problem like this arose, can you do it or can't you do it? And when I was confident and told myself I could do it, then I got through it. Yes, even if I let myself down from time to time. But I got through it. (1. Interview) |
| Simone; 60 | I had migraines for 40 years and had to find my ways. Also finding strategies, actually how to deal with it. Yes, how to beam yourself away. So whenever I've been on vacation, I've always created some images for myself somewhere, eyes closed, ears open, nose sucked in the atmosphere and I can get that out. That works well. Yes, I have to have that, otherwise I'd go crazy. (1. Interview) |
| Ute; 80 | I mean, I've had so many operations in my life that one more or one less doesn't really matter to me. But now a third one, that has mobilized all my strength. (1. Interview) |

**Table 8. Uncertainty and unpredictability.**

| | |
|---|---|
| Chloe; 80 | He (physician) recently saw an X-ray and an MRI and he said: "Everything's great!" But yes, it's not like that! Maybe not quite like that. (1. Interview) |
| Dominik; 70 | I was terrified on the day of the operation and especially when I woke up, whether 1. the leg was still attached and 2. whether this one-sided change worked, and it did. (2. Interview) |
| Edgar; 62 | I think that if you ask other patients, the time at which they decide to take such a step (revision) will vary. One person suffers for a short time and decides to do it, another suffers for longer and still takes time. It's just a question of how do I cope with the pain? How long do I suffer? One person says: "Well, I'm not going to put up with it anymore." And the other says: "Maybe it will get better at some point." So that's what I think. This pivotal point. Yes? Along the lines of: "I can do it somehow on my own or, now I have to do it (revision surgery)." (1. Interview) |
| Frida; 80 | So, this operation. I thought, or we all thought, that it would go well. And so did Dr X. The results were good and that's why the operation was performed. And I was supposed to get a completely new knee. But that didn't happen. I don't know why. Dr X didn't operate any further. He put cement in the knee. (2. Interview) |
| Irene; 63 | What will happen next to me, is totally unpredictable. (2. Interview) |
| Jan; 73 | Yes, that's when I got a fright. Well, that it was that bad. Then the fistula has burst open in such a short time in November and I was already here for a revision at the beginning of December. (1. Interview) |
| Karl; 57 | Yes, in the end it was this surgery here that showed us what was actually going on (referring to the extent of the infection). (1. Interview) |
| Mona; 71 | That's my situation, that I don't know what to do next. And whether it will help me at all. And my situation now is that the examination that was carried out in November and the statements that were made there do not match what has now been said. And I'm a bit unsure about that. (2. Interview) |
| Simone; 60 | Back then he was saying: "Yes, we're taking the hip out now, Mrs Simone." And when I looked at him and said "How?" I said "Doctor, stop here! Can you please explain that to me?" (1. Interview) |
| Ute; 80 | So, I went to my surgeon for the first time after 4 weeks. He stroked my wound once and said: "I'll give you another week" and then later he said: "I don't think it will get any better, we'll have to do the third operation." That almost knocked me off my feet and I said "Yes, but why?" (1. Interview) |

**Table 9. 'In-Between State'.**

| | |
|---|---|
| Berta; 81 | So, I've already expressed a great thing and said: "I have the feeling I'm wavering between suicide and hope ((laughs))." (1. Interview) |
| Dominik; 70 | I still want to have quality of life and with the right leg (TJA of the hip), which went well, I got quality of life. With the left one (side with the PJI), I was bombed back to a level of quality that was even worse than before. (2. Interview) |
| Frida; 80 | I'm a bit in the process of coming to terms with it. Although, I actually cannot accept this circumstance in my life. It's so hard because I can no longer do so many things that I used to love doing. (2. Interview) |
| Irene; 63 | If it stagnates somewhere. That's the problem, that's the problem. So in principle, of course, I fell completely into a hole, because I thought everything would work out at some point. (1. Interview) |
| Mona; 71 | My condition has remained the same. My emotional state has neither worsened nor improved and, as I said, neither has my health. So I'm kind of stuck in the middle. (2. Interview) |
| Tina; 82 | Let me tell you, you can't give yourself up. You can't give up. It always comes back. But as I said, I really try to make the best of it. I go to physiotherapy and always try and hope. What can I do but hope. That it gets a little better. But yesterday I was really knocked out again. That's because I couldn't crawl at all. So, it's strange. (2. Interview) |

results in a sensation of 'hanging in there' or 'holding on to it', where moments of perceived progress are frequently followed by regression, and vice versa. In daily life, this ebb and flow leads to emotional fluctuations, wavering between optimism or hope, and resignation or despair. Thus, patients also oscillate between accepting and denying their current status, ultimately exemplifying the internal struggle of living with chronic PJI. Consider Table 9 for illustrative excerpts:

**3) Unstable patient-physician relationships.** Many patients reported experiencing ineffective or even incorrect treatments, which severely undermined their trust in healthcare providers and their sense of being in 'good hands' (see Table 10). This rupture of trust not only undermines patients confidence in medical care but also impacts their decision-making regarding treatment, which in turn can ultimately influence treatment outcomes after PJI.

Throughout our data analysis, we found that patients often reported periods of losing trust in their physicians, followed by moments of rebuilding that trust later in their treatment journey. Trust emerged as a central component of the patient-physician relationship, particularly in the long and intricate healing and treatment processes associated with PJI, where both patients and physicians can be faced with challenging treatment trajectories. The complexity and extended duration of these treatment journeys can increase the risk for disruptions in the patient-physician relationship, making it essential to actively work on rebuilding trust later as treatment progresses. Consider Table 11 for illustrative quotations:

Throughout the data, patients frequently shared feelings of not being taken seriously by healthcare providers and not being adequately informed about the next steps in their treatment. Some expressed concerns that they were viewed merely as an "interesting" case or condition, rather than being seen as whole individuals with the full range of experiences and suffering related to the disease. Consider Table 12 for excerpts:

Taken together, these excerpts encapsulate the central aspects of the patient-physician relationship that are particularly distinctive in the treatment journey of PJI patients. These factors include the dynamics of trust, characterized by its loss, perceived treatment failures, and eventual restoration of trust as treatment progresses. Conversely, adopting a holistic approach —acknowledging patient concerns and providing detailed information— can be crucial in fostering a positive patient-physician relationship.

## Discussion

Our qualitative interview study explored the multifaceted nature of coping with chronic PJI within the context of treatment and healing. We found that patients develop a process-oriented mindset, which plays a central role in managing both the physical and emotional burdens of the disease. This approach involves a continuous adaptation to daily challenges and losses, active expectation management by shifting the focus from medical to personal goals, and the development of self-efficacy —a self-directed process through which patients take control of their treatment and healing journey. These coping strategies are shaped by the fluctuating nature of treatment circumstances and trajectories that these patients face. This encompasses navigating the uncertainty and unpredictability of their illness, dealing with the strain of unstable patient-physician relationships, and enduring what we have termed an 'In-Between State', where patients oscillate between hope and despair, progression and regression of their perceived health status. These challenges create a complex environment for PJI

**Table 10. Ineffective treatment.**

| Berta; 81 | This is completely unsatisfactory (treatment at the time) and also with bad results. It could have been completely different if he hadn't operated stupidly on me all the time. He must have seen the inflammation values! He must have realised that it couldn't be operated on. He should have just done something with antibiotics. No, he operated on me three times and then so badly that I got the thing stiffened. That's basically the end of the story, isn't it? It didn't have to be that way. (1. Interview) |
|---|---|
| Dominik; 70 | Of course, you're unsure at first when you say: "Man, he messed you up." And I say that without accusation. (1. Interview) |
| Georg; 55 | It was quite a struggle to get here. Until my doctor realized that he couldn't get any further.<br>(1. Interview) |
| Jan; 73 | And then my sister-in-law took me up to the hospital on Sunday and the doctor wanted to tell me, because the inflammation value was 100, let's just say I don't remember, quite high anyway … they called me and said they couldn't admit me now. (1. Interview) |
| Mona; 71 | For example, because that's where you start with this operation. My orthopaedic surgeon who operated on me. After I came back from rehabilitation and the blood values were very suspicious, he did another puncture and then the infection was detected. And then he said well, we'll have to wait and see. (1. Interview) |
| Ute; 80 | Because the doctors, at least back then, the famous Prof X and also Dr Y, they did everything they could to suggest that I am to blame (for the infection) and I just think that's cheeky.<br>(1. Interview) |

**Table 11. Loss and gain of trust.**

| Loss of trust | |
|---|---|
| Berta; 81 | Well, that you feel you're in good hands, right? And that hasn't been the case so far. Because he was standing there when the broth (pus from the fistula) was running, he was standing there and laughing. "It's good that it's running out and oh, that will work out…" and in the same manner of speech, yes. That's not an exaggeration. (1. Interview) |
| Chloe; 80 | I say: "You know, I may be old, but I'm not senile in the head yet. Yes, I know exactly what you said!" And then of course I didn't go back, because I didn't have any trust in him. Once again the feeling that he didn't believe me the whole time. And this smug way of leaning back in the chair and saying: "So, what am I supposed to do with you now? I don't even know what you want from me." (1. Interview) |
| Frida; 80 | That is all. This is to say that Dr X did not continue with the operation. I will never forget that. And also that he never tried to give me any advice. I was simply thrown into the water after the operation. Now swim or don't! (2. Interview) |
| Georg; 55 | "Why don't you know that (date of the revision)?" I said: "You're the chief physician. You must know that. I've been back here since Thursday. You're the chief physician because you can do it. Not because you won the dice roll." (1. Interview) |
| Mona; 71 | I see through people very quickly. And sometimes I get the impression from doctors that they think patients are a bit stupid, that they don't realise what is going on. (1. Interview) |
| Simone; 60 | „Yes, you have an infection there." And then, BAM, BAM, BAM (revision)! And there I shut myself off. You can't just throw me a lump and say "Eat it!". I can't, no. And then, as I said, I tried for days, but I said to my husband "I can't! He threw my trust out of the window with that statement." That's broken, it won't work. I can't do that. I can't relax on the table and have an operation. I can't do it. (1. Interview) |
| Ute; 80 | "Well, it (TJA of the hip) all has to come out again, the infection is still in there!" And then I went home and said: "I definitely do not want to go back to that hospital, although I think the surgeon is a nice guy, but that wasn't good for me, the situation wasn't good." (1. Interview) |
| Gain of trust | |
| Berta; 81 | This is the first time that a doctor has spoken to us (reference to husband) and we are always amazed. We are always amazed when we come here so that we say: "Doctors take their time here." And even if we have to wait, we don't care, we know that they also take their time with the others. And then he takes his time with us too. That's just nice and pleasant. (1. Interview) |
| Chloe; 80 | It makes a big difference if I have to go to the doctor again, where I have no confidence at all. Here I tell myself that I have to go to and he'll know what he's doing. And I have absolute trust in that. (1. Interview) |
| Dominik; 70 | So, I would definitely give other patients the advice that they shouldn't be afraid of an operation if it's necessary, if the doctors think that it needs to be treated as soon as possible, then they should have it done right away. (2. Interview) |
| Edgar; 62 | Yes, and he says: "Of course it's a difficult operation." Then I asked him: "Listen to me. How many times have you done this?" He says: "Probably 50 or 70 times." Then I say: "Then that's good." Yes. Of course I can't verify that, but I can trust him. (1. Interview) |
| Georg; 55 | I got to know some great doctors. Then I stuck to the treatment and participated, and I really enjoyed it. (1. Interview) |

patients, necessitating the development of coping strategies that reflect the personal resilience required to manage chronic PJI.

## Drawing a bigger picture: contextualizing process-orientation and fluctuating treatment circumstances

Process-orientation refers to continuous adaptation to daily challenges arising from physical constraints and losses of routines, relationships, activities, living arrangements and participation in public life. Patients perceived these challenging circumstances as a disconnection from 'normal' life, leaving them uncertain about whether they could ever regain their pre-infection lifestyle. When faced with potential long-term disability, preserving self-esteem and functionality becomes crucial for adapting to PJI. This can involve finding ways to navigate day-to-day life, such as managing household tasks with walking aids or participating in leisure activities despite mobility restrictions or pain. Small changes in routines —like memorizing tripping hazards at home or accepting help from others— can have a significant impact for patients' well-being.

A second facet of process-orientation is characterized by active expectation management, where patients redirect their focus from medical to personal goals. This approach reflects a deliberate shift away from the hope of a complete cure, instead placing emphasis on living a fulfilled present life by reintegrating personally meaningful and self-defining aspects that were overshadowed by the PJI (e.g., conducting a choir from a wheelchair or preserving a small

**Table 12. Patient-physician relationship shaping factors.**

| **Missing holistic perspective** | |
|---|---|
| Berta; 81 | But this emotional side falls by the wayside because the doctors don't have time. Neither do the nurses. Everyone is annoyed. What can you call it? It was simply degrading and inhumane somewhere. It wasn't what you'd expect when a sick person wants treatment. Or yes, wants to be listened to with their grief. (1. Interview) |
| Dominik; 70 | You were left alone like that and I thought that was very, very, very bad, because I have to say it was dramatic. It was a dramatic situation. I've never experienced such anxiety in my life and I have to say that I didn't like the indifferent way I was treated. (1. Interview) |
| Mona; 71 | They treated me like they were somehow standing next to a machine and they have to fix it somehow. Yes, I don't feel like was being treated as a human being. (1. Interview) |
| Nico; 61 | Yes, I think they could have asked me at the time if they could do something good for me. But my condition… Not again. "Yes, Mr Nico, you've had quite a few operations." So I got chewed up with what I already knew and then, painkillers. (1. Interview) |
| Simone; 60 | I spent a good three weeks in hospital, and it really wasn't all that easy, so I think it's really appropriate that something happens. It's not just an operation, it does something to you.<br>(1. Interview) |
| **Not being taken seriously** | |
| Berta; 81 | The first TJA that was implanted was already infected and that was the pain I always had. The bone was so damaged, and nobody noticed. I kept saying: "It hurts so much." Can't you measure the bone density if someone keeps complaining? So that you can maybe see that something is degrading? No one looked in there. It was always like: "Well, she's got nothing." (1. Interview) |
| Chloe; 80 | Yes, yes. So, the fact that things have gone wrong in the meantime. I say, well, things could have gone differently if I had been believed. (1. Interview) |
| Dominik; 70 | But he always gave me the feeling: "It'll be fine, you just have to wait." (1. Interview) |
| Edgar; 62 | In retrospect, I think it's a bit regrettable, as I've already mentioned. The surgeon at hospital X, after the wound hadn't closed for weeks and months, didn't take action. That has nothing to do with accusing him or anything else, but from my point of view he should have said: "Listen, something's not going right, yes. It's not healing properly here, no. We need to really, really get to grips with it." (1. Interview) |
| Nico; 61 | Well, let me tell you, back then nobody told me, or nobody realized that it was an infected endoprosthesis. But five doctors looked at me and said: "Yes, I've already told you that this is basic pain from the operation, Mr. Nico. No, there's nothing to worry about." So, and it got worse and worse. (1. Interview) |
| Simone; 60 | But also the knowledge that you weren't being listened to. Of course, everything possible was done here to smooth things over. But despite all that, I still have to endure it. It is and remains in my memory, this infection. You don't know, could it have been prevented or not? And that is annoying, it's really annoying. (1. Interview) |
| **Deficient informedness** | |
| Berta; 81 | Well, if we (reference to husband) were a bit more versed, we'd write a book about it, as a warning for other people who might be in the same situation than us. (2. Interview) |
| Dominik; 70 | She (treating physician) explains things to me in great detail. She really has the calm, the nerve to listen to my stupid questions all the time and say: "Yes, that's the way it is and that's the way it is and that's what we do." I personally think that's great, that you're taken along with your healing process. And I think that's very important, at least for some of the patients. And not just lying in your room and people are running past you outside in the corridor. And you think: "What is it now?" (2. Interview) |
| Mona; 71 | And then I was completely in the dark and didn't know at all what to expect. I mean, well, maybe there are people who are very sensitive when you tell them the whole truth. But I'm not like that. (2. Interview) |
| Simone; 60 | Then you (physician that initially treated her) just have to make an effort the next day and say: "Let's try to get to the bottom of it again today.", and not always assume that the patient has the doctor's knowledge. (1. Interview) |
| Ute; 80 | It was just a question of being informed, because I don't think it's okay if someone doesn't inform me. (1. Interview) |

involvement in family gardening projects). Patients thereby develop a sense of acceptance for current restrictions, creating space for further adjustments while recognizing and leveraging personal resources rather than focusing on deficits, regardless of their severity.

Lastly, the third aspect of process-orientation centers around self-efficacy as a key resource for navigating the treatment and healing process in a self-determined manner. Patients foster this by proactively regaining control over their personal and medical decisions. They often addressed challenging situations during treatment by actively taking charge, whether by demanding information or changing physicians after ineffective treatment, thereby directly influencing their outcomes. In doing so, they not only develop but also preserve a significant sense of self-efficacy. Furthermore, patients managed to mobilize strengths from critical past

experiences, that had been successfully overcome, and therefore took on an active role in their current treatment and healing journey.

These coping processes are deeply influenced by the unique treatment context of PJI. Along this journey, patients frequently face complex decisions, such as deciding in favor or against another revision surgery, that exceed their knowledge and experience as laypersons. In a field where the outcome of treatment decisions is often unclear and recommendations vary among healthcare practitioners, patients experience a profound sense of unpredictability and uncertainty regarding their care. This uncertainty leads patients to oscillate between phases of perceived progress and regression, creating what we term an 'In-Between State'—a condition that is characterized by fluctuation feelings and swinging between optimism and resignation.

Patients also reported instances of ineffective or incorrect treatment, which further undermined their trust in healthcare providers and disrupted the patient-physician relationship. Trust eroded further when patients felt that their symptoms were dismissed or when they lacked clear information concerning their future prospects. Conversely, patients with PJI benefit from a holistic, patient-centered approach where transparency and empathy are prioritized. This fosters the rebuilding and maintenance of trust, which is central to a positive patient-physician relationship [16,17].

Our study reveals a novel perspective on the fluctuating trust between patients with PJI and their physicians, highlighting the instability of patient-physician relationships throughout the treatment process. This finding is particularly relevant in the context of evolving health care systems in developed countries, where patients are increasingly viewed as autonomous actors. In such systems, treatments are no longer perceived as a one-size-fits-all solution, but rather as individualized journeys, significantly influenced by the quality of the patient-physician relationship [16]. Trust plays a pivotal role in patient decision-making, especially when facing complex decisions such as whether to proceed with additional operations. This process is heavily influenced by patients' trust in the effectiveness of treatment and confidence in the healthcare providers [17–19]. The frequent experiences of ineffective treatments among our cohort underscore the critical and fragile nature of trust, as its loss and subsequent efforts to regain it become central to the PJI treatment experiences. This aligns with existing knowledge, which emphasizes the significant impact of trust on the health of chronic patients, noting that these communication-based effects are not facilitated by a single clinician-patient encounter but rather reflect cumulative experiences over time [20].

Our findings also underscore the importance of a process-oriented mindset in managing PJI, as proposed by existing coping theories [21] that emphasize the importance of assessing one's own resources adequately and adapting actively when coping with chronic illnesses [22, 23]. Thus, a process-oriented approach shifts the focus from the often unpredictable long-term goal of complete recovery to more immediate, manageable objectives that patients can influence directly. This differs significantly from the traditional outcome-focused model prevalent in somatic medicine, where success is measured by the resolution of symptoms or reaching pre-illness functioning. Instead, our study highlights how a process-oriented approach can mitigate feelings of powerlessness, since regaining pre-infectious outcomes is not always possible [9]. By promoting a more active, self-efficacious stance among patients, this approach enhances their immediate quality of life.

## Implications for research and practice

To enhance the integration of our findings into clinical practice, healthcare providers should consider conveying a process-oriented perspective on healing that emphasizes ongoing adaptation and active patient involvement. Encouraging patients to set personal, manageable goals

and supporting them in regaining a sense of control can improve their overall engagement in treatment and healing. Additionally, fostering transparent communication and trust between patients and physicians is crucial, particularly in managing expectations and addressing the fluctuating nature of PJI recovery. Transparent communication plays a pivotal role in building this trust and ensuring that care is truly patient-centered [20,21]. Implementing these strategies could significantly enhance patient care by aligning treatment with patients' physical and psychological needs, ultimately leading to better health outcomes [18].

The psychological impact of PJI has been linked to the experiences of oncology patients, as highlighted by Knebel and colleagues [3]. Since the 1970s, Psycho-Oncology has become a firmly integrated part of oncological care, providing support to patients, survivors and relatives through networks like the International Psycho-Oncology Society [24]. Considering the conclusive research on the mental burden of PJI [3,6–8] and the void of strategies for addressing it —an issue that extends to infectious musculoskeletal diseases in general [7]— we propose the establishment of a new subdiscipline within endoprosthetics: 'Psycho-Endoprosthetics'. A structured approach would pave the road for the creation of national and international networks focused on research and practice, laying the groundwork for essential systematic studies and the integration of tailored psychosocial support for PJI patients.

## Limitations

While our study offers valuable insights, several limitations must be considered. Conducted within a single center, the findings may have limited transferability to other clinical settings. Furthermore, the context in which the interviews were conducted —within the same department responsible for patients' treatment— may have introduced a tendency towards social desirability bias, with patients potentially framing their responses to be favorable to their caregivers. In qualitative research, contextualization is crucial, and results are deeply intertwined with specific environments, which limits broad generalization. Nevertheless, the study achieved thematic saturation, and the rigorous application of triple and double coding procedures enhanced the robustness of the analysis. The follow-up interviews, which tracked patients' developments and experiences over six months, also added depth to the data. We acknowledge potential sources of bias inherent in our study design, particularly the risk of recall bias, given the lengthy treatment journeys of the patients, which may have affected their ability to accurately remember past events. However, the longitudinal approach used is particularly well-suited to explore the complex healing journey in chronic PJI treatment. We are cognizant that due to our focus on chronic cases, our sample exhibits great variability regarding treatment stages, the number of revision surgeries and the overall duration of therapy. While such variability might be seen as a limitation in positivistic research, where the minimization of diverging interindividual variables tends to be the goal, it is considered a strength in qualitative study designs. This diversity enabled us to explore the wide range of experiences and perspectives that accompany PJI, enriching our study's findings. The study's strengths suggest that despite the context-specific nature of our findings, they may provide transferable insights for other long-term care contexts.

## Conclusions

From the patient's standpoint, PJIs lead to irreversible life change that significantly affect their perceived quality of life and well-being independent of their infection status [7,25]. While the necessity of psychological support for PJI patients has been voiced before [6,9] and is strongly encouraged by our findings, there is currently no established framework for delivering this

care. As a result, the responsibility for providing psychosocial guidance falls largely on physicians. Therefore, we strongly recommend that practitioners promote a process-oriented mindset early in the treatment process. This approach should be initiated right from the start of treatment in a preventive manner and maintained throughout follow-up appointments, whether the patient is still undergoing active treatment or not. Understanding that PJI as an illness extends beyond the treatment period —with the possibility of both physiological and psychological challenges persisting despite the achievement of medical goals— can sensibilize partitioners for the holistic healthcare needs of PJI patients.

Our data suggests that patients, whether at the beginning of their treatment journey or after being cured of the infection, should have the opportunity to address the impact of PJI on their lives. As the psychological toll of PJI requires broader recognition, we propose the establishment of a new subdiscipline, namely 'Psycho-Endoprosthetics'. This initiative would facilitate more effective dissemination of knowledge and interdisciplinary collaboration in the field of PJI. Thereby addressing the critical need to incorporate mental health professionals into the interprofessional team for PJI care.

## Supporting information

**S1 File. Interview guide for first and follow-up interviews.**
(DOCX)

**S2 Fig. Initial Thematic Map.**
(TIF)

**S3 Table. COREQ checklist.**
(DOCX)

**S4 Fig. Coding tree.**
(TIF)

## Acknowledgements

We are deeply grateful to the patients for their time, openness, and willingness to participate in this study. We would also like to express our gratitude to Lea Schulte for her invaluable support and assistance in conducting this research. Her expertise and dedication have significantly contributed to the success of this project. Additionally, we extend our appreciation to the team of the septic surgery department for their essential contributions and collaboration.

ChatGPT-4 was utilized during the review process to refine the original text, for which the authors take full responsibility.

## Author contributions

**Conceptualization:** Vincent Tilo Krenn, Andrej Trampuz, Carsten Perka, Sebastian Meller.

**Data curation:** Vincent Tilo Krenn.

**Formal analysis:** Vincent Tilo Krenn, Maria Sarah Bönigk, Martin Liebisch.

**Investigation:** Vincent Tilo Krenn.

**Methodology:** Vincent Tilo Krenn, Maria Sarah Bönigk.

**Project administration:** Vincent Tilo Krenn, Andrej Trampuz, Carsten Perka, Sebastian Meller.

**Validation:** Vincent Tilo Krenn, Maria Sarah Bönigk, Martin Liebisch.

**Writing – original draft:** Vincent Tilo Krenn, Maria Sarah Bönigk, Andrej Trampuz, Martin Liebisch, Carsten Perka, Sebastian Meller.

**Writing – review & editing:** Vincent Tilo Krenn, Maria Sarah Bönigk, Andrej Trampuz, Martin Liebisch, Carsten Perka, Sebastian Meller.

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
