## [Decision Letter · Decision Letter 0]

14 Aug 2024

PONE-D-24-21025Coping with chronic Periprosthetic Joint Infection: A Qualitative Study on Patient Experiences in Treatment and HealingPLOS ONE

Dear Dr. Krenn,

Thank you for submitting your manuscript to PLOS ONE. After careful consideration, we feel that it has merit but does not fully meet PLOS ONE’s publication criteria as it currently stands. Therefore, we invite you to submit a revised version of the manuscript that addresses the points raised during the review process.

We look forward to receiving your revised manuscript.

Kind regards,

Adriana Calderaro

Academic Editor

PLOS ONE

Journal Requirements:

2. Please amend your authorship list in your manuscript file to include authors Dr. Vincent Krenn, Maria Sarah Bönigk, Martin Liebisch, Carsten Perka, Carsten Perka, and Sebastian Meller.

3. We note that your Data Availability Statement is currently as follows: "All relevant data are within the manuscript and its Supporting Information files."

Reviewers' comments:

Reviewer's Responses to Questions

**Comments to the Author**

1. Is the manuscript technically sound, and do the data support the conclusions?

Reviewer #1: Yes

Reviewer #2: Yes

2. Has the statistical analysis been performed appropriately and rigorously? 

Reviewer #1: Yes

Reviewer #2: N/A

3. Have the authors made all data underlying the findings in their manuscript fully available?

Reviewer #1: Yes

Reviewer #2: Yes

4. Is the manuscript presented in an intelligible fashion and written in standard English?

Reviewer #1: Yes

Reviewer #2: Yes

5. Review Comments to the Author

Reviewer #1: Abstract

1. In the objectives of abstract section you need to find a more reasonable rationale; for example, "due to long-term multiple medical and surgical treatments..."

2. It would be appropriate to simplify and shorten the conclusion section for better clarity: 'The physical challenge of dealing with a prolonged infection through various treatment modalities, along with the mental fatigue negatively impacted by additional illnesses, can be managed by creating a supportive environment that encourages step-by-step recovery. "

Introduction

1. Which aspect is unclear? You should conclude the sentence more clearly: '...the post-treatment patient outcomes remain fraught with uncertainty.'"

2. It would be more appropriate if this part is revised to indicate that it is not emphasized much by the treating physicians and is pushed into the background: '.... The psychosocial impact of PJI, especially in chronic cases, is not fully understood.'"

3. What is the duration of the process you are referring to with '... largely overlooking the long-term psychological adjustments...'?"

Material & Methods

This section states that the survey study was properly designed, implemented, and analyzed.

1. However, the two-stage surgical experience of the 18 patients mentioned in Table 1 could be presented more clearly. In normal surgical techniques, patients treated for chronic PJI typically undergo at least 6 weeks of antibiotic therapy, followed by a window period, after which a second-stage treatment is needed if the acute-phase reactants remain stabilized, leading to the start of the rehabilitation process. What factors prevented the infections in the included patients from being treated with the classical approach? Were there any additional diseases, virulent microorganisms, or other reasons that could have affected the survey results? If so, it would be more appropriate to revise the title to reflect this situation."as "Coping with failed revision Total Knee dur to infection: ..."

Both the Results and Discussion sections are difficult to follow and complex. It would be helpful to make them shorter and clearer, particularly by

1. dividing the Discussion section into paragraphs,

2. starting with the answer to the research question,

3. including the proposal of psychoendoprosthetics in the Discussion, and removing it from the Conclusion as "Developing a new sub-discipline, psychoendoprosthetics, to facilitate interdisciplinary collaboration will make it easier to achieve this goal."'

4. Additionally, it would be appropriate to add a paragraph detailing any limitations before the Conclusion.

Congratulations on this well-thought-out study that requires significant effort.

Reviewer #2: The article examines the psychological coping strategies of patients with chronic periprosthetic joint infection (PJI) and provides significant contributions to our knowledge on this topic. This is a subject rarely addressed in the literature, which makes the study highly original.

The study conducted an in-depth examination of patients' experiences using thematic analysis. However, the details of the methods used and the explanation of the analysis process need to be more comprehensive. Specifically, it is important to better explain how the themes used in the coding process were determined and what criteria were used for their selection. The integration of the findings into clinical practice and their potential impact on patient care should be discussed in more detail. The coding tree presents a systematic approach; however, visual and explanatory elements should be added to enhance clarity and transparency.

6. PLOS authors have the option to publish the peer review history of their article (what does this mean? ). If published, this will include your full peer review and any attached files.

**Do you want your identity to be public for this peer review?** For information about this choice, including consent withdrawal, please see our Privacy Policy .

Reviewer #1: **Yes: ** ismail demirkale

Reviewer #2: No

---

## [Author Response · Author response to Decision Letter 1]

28 Aug 2024

Letter to the reviewers:

We would like to express our sincere gratitude to the reviewers and the editor for their valuable comments and suggestions. We have carefully considered each point and have revised our manuscript accordingly.

Response to the comments of reviewer #1:

Abstract:

1. In the objectives of abstract section, you need to find a more reasonable rationale; for example, "due to long-term multiple medical and surgical treatments..."

• We have adapted our rationale to convincingly highlight the Study’s objectives, as was mentioned by Reviewer #1. The sentence is now stated as followed:

o Periprosthetic joint infections (PJI), along with the extensive medical and surgical interventions required for treatment, impose a substantial psychological burden on patients. Given the need for patients to adapt to long-term physical limitations and ongoing medical challenges, this qualitative study aims to explore nature of psychological coping amongst patients with chronic cases of PJI.

2. It would be appropriate to simplify and shorten the conclusion section for better clarity: 'The physical challenge of dealing with a prolonged infection through various treatment modalities, along with the mental fatigue negatively impacted by additional illnesses, can be managed by creating a supportive environment that encourages step-by-step recovery. "

• We acknowledge the need mentioned by Reviewer #1 to shorten and simplify the conclusion. We have revised it, reducing the length from 732 to 436 characters, while ensuring that the key conclusions and implications of our study are clearly presented.

Introduction:

1. Which aspect is unclear? You should conclude the sentence more clearly: '...the post-treatment patient outcomes remain fraught with uncertainty.'"

• In response to the feedback of Reviewer #1 (Introduction 1.), we have clarified the sentence. Please find the reworked version as followed:

o While diagnostic and treatment strategies for PJI have advanced considerably [5], the post-treatment patient outcomes remain uncertain, especially due to the unclear length of recovery and the variability in regaining post-operative quality of life.

2. It would be more appropriate if this part is revised to indicate that it is not emphasized much by the treating physicians and is pushed into the background: '.... The psychosocial impact of PJI, especially in chronic cases, is not fully understood.'"

• Responding to Reviewer #1 (Introduction 2.): We have revised this part of the sentence to clarify this point. Please find the refined version as followed:

o … PJI, particularly in chronic cases, is often underemphasized by treating physicians and remains not fully understood.

3. What is the duration of the process you are referring to with '... largely overlooking the long-term psychological adjustments...'?"

• We have addressed and refined the sentence as suggested by Reviewer #1 (Introduction 3.) to clarify our intended meaning and specify the time span we are referring to. Additionally, we took this opportunity to clearly outline the aims of other studies in the field concerning the time span related to the burden of PJI, thereby making the relevance of our study more apparent. Please find the new version as followed:

o Research has primarily focused on the immediate medical responses and short-term recovery processes of patients who have undergone one- or two stage revision surgery [8,9], largely overlooking the long-term psychological adjustments patients must make in order to integrate physical, mental and social consequences of chronic PJI into their lives —a highly individual process that can span over years and may be prolonged due to reinfection. Further studies on the burden of PJI have examined the implications for mental health, assessing outcome measures like depression, anxiety, and overall psychological distress [3,6].

Material and Methods:

1. However, the two-stage surgical experience of the 18 patients mentioned in Table 1 could be presented more clearly. In normal surgical techniques, patients treated for chronic PJI typically undergo at least 6 weeks of antibiotic therapy, followed by a window period, after which a second-stage treatment is needed if the acute-phase reactants remain stabilized, leading to the start of the rehabilitation process. What factors prevented the infections in the included patients from being treated with the classical approach? Were there any additional diseases, virulent microorganisms, or other reasons that could have affected the survey results?

• Due to the extensive changes made in the Materials and Methods section (and also in the Discussion), we deactivated the tracking mode to enhance readability and to guarantee that “word” runs smoothly. We have included comments in the “Revised Manuscript with Track Changes” to ensure our changes are clearly understood.

• We have further refined and detailed the Methods and Materials section in response to the reviewer's #1 feedback to provide greater clarity and detail regarding the surgical protocols, diagnostic procedures, inclusion/exclusion criteria, and pathogen stratification. Additionally, we described our reinfection management with both EBJIS guidelines and the Delphi consensus criteria. These enhancements ensure a more comprehensive understanding of our methodology and address the specific concerns raised.

• Further, we have adapted Table 1 to more accurately reflect the individual surgical experiences of the patients. We specifically addressed Reviewer #1's concerns regarding virulent microorganisms and the factors that prevented treatment with the classical approach. While we greatly appreciate the suggestion to include comorbidities, we chose not to list them at this point. Although comorbidities do influence treatment strategies, they are not directly related to our research question. This approach aligns with similar studies (Moore et al. 2015; Palmer et al. 2020), which have also refrained from listing comorbidities for this reason. We hope that our reasoning is understandable and that the changes clarify the attributes of our cohort.

If so, it would be more appropriate to revise the title to reflect this situation."as "Coping with failed revision Total Knee dur to infection: ..."

• We have considered Reviewers#1 feedback and adapted the title to better clarify the study's aims.

o The new title is as followed: “Coping with chronic periprosthetic joint infection after failed revision of total knee and hip arthroplasty: a qualitative study on patient’s experiences in treatment and healing”

Results and Discussion:

Both the Results and Discussion sections are difficult to follow and complex.

• We addressed the difficulty of following our initial complex „Results and Discussion“ section, as was mentioned by Reviewer #1 by clarifying our results to make them more comprehensive. Track changes can be found within the revised manuscript with track changes.

It would be helpful to make them shorter and clearer, particularly by

1. dividing the Discussion section into paragraphs,

2. starting with the answer to the research question,

3. including the proposal of psychoendoprosthetics in the Discussion, and removing it from the Conclusion as "Developing a new sub-discipline, psychoendoprosthetics, to facilitate interdisciplinary collaboration will make it easier to achieve this goal."'

4. Additionally, it would be appropriate to add a paragraph detailing any limitations before the Conclusion.

• Since we completely revised the Discussion, we decided to deactivate the tracking option and explain our changes via comment function in the revised manuscript with track changes.

• Our changes meet the remarks of Reviewer #1 by dividing the Discussion into paragraphs and adding sub-headers to enhance the overview.

• As mentioned by Reviewer #1, we started the Discussion by immediately addressing the research question.

• We further included the proposal for ‘Psycho-Endoprosthetics’ in the discussion and detailed the study’ limitations before the Conclusion.

Response to the comments of reviewer #2:

The study conducted an in-depth examination of patients' experiences using thematic analysis.

1. However, the details of the methods used and the explanation of the analysis process need to be more comprehensive. Specifically, it is important to better explain how the themes used in the coding process were determined and what criteria were used for their selection.

• A new text on page 8 of the revised manuscript with track changes was added in response to Reviewer #2's feedback regarding the need for a more detailed and comprehensive explanation of the analysis process. We also emphasized how themes were selected and the criteria applied in this process. Please find the text below:

o The methodology of thematic analysis (TA) provides the opportunity for a relativistic inductively oriented analysis. Grounded in an experiential framework we focused on amplifying the voices of lived experiences. TA captures both semantic and latent meanings and offers a comprehensive approach that includes descriptive and interpretative accounts of the data [14,15]. A six phase TA was undertaken to explore patterns of shared meaning across data. The six phases were applied in a distinct yet recursive manner [14]. The first step of TA involved familiarization with the data by thoroughly reading and re-reading the interviews by researchers VK, ML and MB. In the second step, we performed an inductive, data-driven coding process by assigning codes to one or two-sentence segments from the transcripts, reflecting an exploratory approach. Throughout this process, inductive codes were continually developed until all expressions related to the lived experiences of patients with PJI were identified and labeled. VK, ML, and MB independently coded 20% of the interviews. The codes were then reviewed, compared, and discussed by the team until a consensus was reached. Subsequently, VK and ML coded the remaining 80% of the interviews separately, followed by another round of review, comparison, and discussion to reach agreement. The codes were then printed and preliminarily grouped. In the third phase, all codes were categorized by VK and MB into initial themes that encapsulated the phenomena under investigation. These themes captured significant aspects of the data by portraying patterns of shared meaning embedded within the dataset [14,15]. Phase four involved reviewing and refining the themes, which were performed by VK and MB. By revisiting the themes in relation to the context of the coded sentences provided by the transcripts, an initial thematic map (see S2 Fig) was developed, visually representing the identified themes and their interrelationships. The interview transcripts and coded segments were re-examined to ensure coherence, leading to iterative adaptations of the thematic map. After capturing all phenomena associated with chronic PJI, the focus was narrowed according to the research question. Themes were reassembled and refined within the thematic map, with repeated reviews of the interview transcripts and coded segments to ensure coherence and contextual accuracy. Afterwards, final themes and subthemes were defined and named. Subthemes were identified as key concepts within their respective overarching theme. The refinement of initial themes into final themes was based on several criteria: reflection of shared patterns across the dataset, illumination of the research topic and its contextualization, non-redundancy, sufficient meaningful data to support the theme, presence of a central organizing concept, clear boundaries of the theme (for inclusion and exclusion), coherence of the theme (avoiding overly broad themes). The importance of a theme was determined not only by its prevalence but also by its salience and significance in addressing the research question. The final themes generated in this process represent the outcome of the analysis (see Fig 1). Phase five was undertaken by researchers VK and MB and focused on the refinement, renaming, and redefinition of the final themes to enhance their clarity and informative value. Lastly, phase six encompassed the writing process, which integrated the final themes into a coherent overall narrative and connected our findings to the existing literature [14]. Subsequently, lively and convincing segments from the transcripts were extracted for each theme and subtheme.

2. The integration of the findings into clinical practice and their potential impact on patient care should be discussed in more detail.

• We have addressed Reviewer #2's feedback by providing a more detailed discussion on the integration of our findings into clinical practice. We now dedicate to this aspect a paragraph named: Implications for research and practice. Please find the new paragraph below:

o Implications for research and practice

To enhance the integration of our findings into clinical practice, healthcare providers should consider conveying a process-oriented perspective on healing that emphasizes ongoing adaptation and active patient involvement. Encouraging patients to set personal, manageable goals and supporting them in regaining a sense of control can improve their overall engagement in treatment and healing. Additionally, fostering transparent communication and trust between patients and physicians is crucial, particularly in managing expectations and addressing the fluctuating nature of PJI recovery. Transparent communication plays a pivotal role in building this trust and ensuring that care is truly patient-centered [20,21]. Implementing these strategies could significantly enhance patient care by aligning treatment with patients' physical and psychological needs, ultimately leading to better health outcomes [18].

The psychological impact of PJI has been linked to the experiences of oncology patients, as highlighted by Knebel and colleagues [3]. Since the 1970s, Psycho-Oncology has become a firmly integrated part of oncological care, providing support to patients, survivors and relatives through networks like the International Psycho-Oncology Society [24]. Considering the conclusive research on the mental burden of PJI [3,6–8] and the void of strategies for addressing it —an issue that extends to infectious musculoskeletal diseases in general [7]— we propose the establishment of a new subdiscipline within endoprosthetics: ‘Psycho-Endoprosthetics’. A structured approach would pave the road for the creation of national and international networks focused on research and practice, laying the groundwork for essential systematic studies and the integration of tailored psychosocial support for PJI patient

3. The coding tree presents a systematic approach; however, visual and explanatory elements should be added to enhance clarity and transparency.

• To address Reviewer #2's request for visual and explanatory elements, we included our initial thematic map in the supplementary material (S2), provided a more detailed description of the analytic process (the added text in the materials and methods section), and added our final thematic map (Figure 1) to offer a comprehensive overview of our results. To further enhance the clarity and transparency of our process, we included photographs of our manual analysis to underscore our robust and strategic approach and also a screenshot from our MAXQDA file.

• We have further updated S3: Table, in order to adequately encompass the expertise of our research team.

Changes in the data availability statement:

We were informed to clarify our data availability statement and to include this in the 'Response to Reviewers’. Please find our updated statement below:

Data Availability Statement

The data from this study involve patients with a rare condition (periprosthetic joint infection). This includes potentially identifying and sensitive patient information. Due to the detailed medical histories disclosed in the study, there is an increased risk of re-identification. Anonymized data from this study will be made available to bona fide researchers after approval from the Ethics Committee of Charité. This data will be handed over to the Research Data Mana

---

## [Decision Letter · Decision Letter 1]

4 Feb 2025

Coping with chronic periprosthetic joint infection after failed revision of total knee and hip arthroplasty: a qualitative study on patients' experiences in treatment and healing

PONE-D-24-21025R1

Dear Dr. Krenn,

We’re pleased to inform you that your manuscript has been judged scientifically suitable for publication and will be formally accepted for publication once it meets all outstanding technical requirements.

Kind regards,

Adriana Calderaro

Academic Editor

PLOS ONE

Additional Editor Comments (optional):

Reviewers' comments:

Reviewer's Responses to Questions

**Comments to the Author**

1. If the authors have adequately addressed your comments raised in a previous round of review and you feel that this manuscript is now acceptable for publication, you may indicate that here to bypass the “Comments to the Author” section, enter your conflict of interest statement in the “Confidential to Editor” section, and submit your "Accept" recommendation.

Reviewer #1: All comments have been addressed

Reviewer #3: All comments have been addressed

Reviewer #4: All comments have been addressed

2. Is the manuscript technically sound, and do the data support the conclusions?

Reviewer #1: Yes

Reviewer #3: Yes

Reviewer #4: Yes

3. Has the statistical analysis been performed appropriately and rigorously? 

Reviewer #1: Yes

Reviewer #3: Yes

Reviewer #4: Yes

4. Have the authors made all data underlying the findings in their manuscript fully available?

Reviewer #1: (No Response)

Reviewer #3: Yes

Reviewer #4: Yes

5. Is the manuscript presented in an intelligible fashion and written in standard English?

Reviewer #1: Yes

Reviewer #3: Yes

Reviewer #4: Yes

6. Review Comments to the Author

Reviewer #1: I would like to thank the authors for their appropriate revisions based on all the suggestions I provided.

Reviewer #3: This is a nicely perfomred study pointing out the psychologial burden of patients suffering PJIs. In my eyes all weaknesses have been addressed appropriately so that I recommend publication in Plos One.

Reviewer #4: 1. really important for the abstract to even make clearer that these are not just patients w/ PJI but ones who failed a prior treatment attempt. this is a really challenging clinical problem.

2. i'm not sure we need a new field but I think this underscores that periprosthetic infection centers should likely have a dedicated psych person on the team.

3. The idea that we have a process-oriented approach to patients is a tough recommendation. I am reluctant for us to put too much additional burden on the surgeons managing these cases. I think everyone tries to do their best. this situation is not just difficult for patients but also for surgeons.

7. PLOS authors have the option to publish the peer review history of their article (what does this mean? ). If published, this will include your full peer review and any attached files.

**Do you want your identity to be public for this peer review?** For information about this choice, including consent withdrawal, please see our Privacy Policy .

Reviewer #1: **Yes: ** ismail demirkale

Reviewer #3: **Yes: ** Malte Ohlmeier

Reviewer #4: No

---

## [Editor Report · Acceptance letter]

PONE-D-24-21025R1

PLOS ONE

Dear Dr. Krenn,

I'm pleased to inform you that your manuscript has been deemed suitable for publication in PLOS ONE. Congratulations! Your manuscript is now being handed over to our production team.

Kind regards,

on behalf of

MD, PhD, Full Professor Adriana Calderaro

Academic Editor

PLOS ONE